# CubeSat Observation of the Radiation Field of the South Atlantic Anomaly

**Pavel Kovář [1,\*]** and **Marek Sommer [2,3]**

1   Faculty of Electrical Engineering, Czech Technical University in Prague, Technická 1902/2, 166 27 Prague, Czech Republic

2   Nuclear Physics Institute of the Czech Academy of Sciences, Husinec-Řež 130, 250 68 Řež, Czech Republic; sommer@ujf.cas.cz

3   Faculty of Nuclear Sciences and Physical Engineering, Czech Technical University in Prague, V Holešovičkách 2, 180 00 Prague, Czech Republic

\*   Correspondence: kovar@fel.cvut.cz

**Abstract:** The movement of the South Atlantic Anomaly has been observed since the end of the last century by many spacecrafts equipped with various types of radiation detectors. All satellites that have observed the drift of the South Atlantic Anomaly have been exclusively large missions with heavy payload equipment. With the recent rapid progression of CubeSats, it can be expected that the routine monitoring of the South Atlantic Anomaly will be taken over by CubeSats in the future. We present one-and-a-half years of observations of the South Atlantic Anomaly radiation field measured by a CubeSat in polar orbit with an elevation of 540 km. The position is calculated by an improved centroid method that takes into account the area of the grid. The dataset consists of eight campaigns measured at different times, each with a length of 22 orbits (~2000 min). The radiation data were combined with GPS position data. We detected westward movement at 0.33°/year and southward movement at 0.25°/year. The position of the fluence maximum featured higher scatter than the centroid position.

**Keywords:** South Atlantic Anomaly; radiation measurement; CubeSat observation; radiation belts



## 1. Introduction

The Earth has two van Allen radiation belts, i.e., the inner and outer, which consist of trapped high-energy charged particles. Most of the particles within the radiation belts originate from solar winds and galactic cosmic rays [1].

The approximately dipolar magnetic field at low altitudes generated by Earth's core is tilted and shifted from the geographic pole. This creates a region with reduced magnetic field intensity, namely the South Atlantic Anomaly (SAA), which is located approximately on the eastern coast of Brazil. The SAA is a place where the radiation of the inner van Allen belt approaches the closest to Earth [2,3]. One of the manifestations of the SAA is the enhanced count rate of protons and electrons coming from the inner radiation belt [4].

The SAA region presents a threat to low Earth-orbiting satellites due to the high probability of single event upsets, failure of microelectronics and premature ageing, as well as to astronauts and their health [5–7]. Therefore, the region has been thoroughly investigated since its discovery. Many of the published papers have focused on the dynamics of the SAA, mainly the variation in its intensity over time and its drift, to predict its future movement. It has been observed that the SAA moves steadily in a northwest direction. This movement has been registered by measurement of the magnetic field [8,9] and by radiation [2,3,10–21]. Table 1 summarizes the publications that have focused on examining the drift of the radiation center of the SAA. As shown in previous studies, a large spread of drift velocities has been reported. Such large discrepancies could be explained by the dependence of the particle flux on the altitude and on particle energy, as described in [2], as many measurements

were performed on different orbits with different instrumentation. Other phenomena that influence the drift of the SAA and its velocity are sun modulation and "geomagnetic jerks". The effect of the solar cycle on drift velocity was described in [3,20,21]. The authors of [19] theorized that the rapid short-term change in the drift of the SAA was caused by a "geomagnetic jerk", which was reported in 2003 by [22]. Geomagnetic jerks occur when the secular acceleration of the magnetic field rapidly changes.

**Table 1.** Summarized results of measured South Atlantic Anomaly (SAA) drift rates.

| Study | Westward Drift (°/Year) | Northward Drift (°/Year) | Altitude (km) | Inclination (°) | Time (Year) |
|---|---|---|---|---|---|
| Konradi 1994 [14] | 0.32 | - | 617<br>450<br>287<br>565 | 28.5<br>28.5<br>39<br>58 | 1990–1991 |
| Badhwar 1997 [15] | 0.28 ± 0.03 | 0.08 ± 0.03 | 438<br>393 | 50<br>51.65 | 1973, 1995 |
| Bühler 2002 [16] | 0.06 ± 0.05 | 0.06 ± 0.05 | 400 | 52 | 1994–1996 |
| Ginet 2006 [17] | 0.43 ± 0.13 | - | 410–1710 | 69 | 2000–2006 continuous measurement |
| Grigorian 2008 [18] | 0.1–1.0 | 0.1 | 307–393<br>220<br>400<br>500–2500<br>400<br>450 | 65<br>81.6<br>51.6<br>81.3<br>51.6<br>51.6 | 1960–2003 |
| Fürst 2009 [19] | 0.248 | - | 592 in 1996<br>488 in 2007 | 23 | 1996–2007 continuous measurement |
| Casadio and Arino 2011 [20] | 0.24 | 0.08 | 782–785<br>780<br>800 | 98.52<br>98.5<br>98.55 | 1991–2010 continuous measurement |
| Qin 2014 [21] | 0.3 | 0.09 | 813<br>833<br>804<br>833 | 98.7<br>98.6<br>98.5<br>98.7 | 1980–2010 Almost continuous measurement |
| Schaefer 2016 [10] | 0.36 ± 0.06 | 0.16 ± 0.09 | 840–860 | 99 | 2004–2013 continuous measurement |
| Jones 2017 [3] | 0.20 ± 0.04 | –0.11 ± 0.01 | 400–600 | | 1993–2011 continuous measurement |
| Ye 2017 [2] | Various (depends on proton energy) | Various (depends on proton energy) | 512–687 | 81.7 | 1994–2007 continuous measurement |
| Anderson 2018 [12] | 0.277 ± 0.008 | 0.064 ± 0.008 | DMSP F8–F18 | DMSP F8–F18 | 1987–2015 continuous measurement |
| Aubry 2020 [13] | 0.639<br>0.329<br>0.256 | –<br>–<br>– | 715<br>1336<br>850 | 98<br>66<br>98 | 2000–2018 continuous measurement |

Another systematic error in the evaluation of SAA drift can be caused by different methods of data processing. One of the most deployed methods for evaluation of the SAA position is Gaussian or Weibull fitting of measured maxima data (flux, dose, and dose rates) over a period of time and calculation of the SAA position based on the fit maximum [3,14–17,21]. In recent years, a new interpolation method based on the calculation of the SAA centroid has been

presented [12,13]. The shortcoming of the centroid method used in [12,13] is that it does not consider cosine-latitude effects.

CubeSats have been used for many scientific studies in recent years. Although CubeSat missions cannot fully replace mainstream space missions, they have proven to be a useful and inexpensive platform for small payloads [23,24]. CubeSats have been used in scientific fields of Earth science [25–27], space weather [28–30], and astrophysics [27,31].

Our 1U CubeSat Lucky 7 was launched with the primary objective of testing communication systems and global positioning systems (GPSs). The secondary objective was to study the ionizing radiation field with two devices, i.e., a piDOSE radiation detector and a silicon diode spectrometer. Due to the failure of the spectrometer, only data from piDOSE were obtained.

The objective of this study is to prove that 1U CubeSats can be exploited for the long-term monitoring of the SAA movement, as all measurements introduced in Table 1 were done on professional large satellites. Moreover, this study focuses on the improvement of the centroid method used for the localization of the SAA center.

In our study, we present a modest dataset of processed SAA location data gathered by the radiation detector piDOSE flown on the Lucky 7 satellite. The SAA location was derived from the radiation data by the centroid method normalized for cosine-latitude effects. We show that the centroid method calculates the SAA location with much higher accuracy than the maxima fitting method. Hence, fewer data are needed to evaluate the SAA location when the centroid method is used. Such features might be crucial in the case of 1U CubeSat, which has very limited power, and broadcasting resources and data transfer to Earth might be problematic. We believe that lightweight, cost-effective CubeSats equipped with radiation detectors such as piDOSE can be utilized for the continuous monitoring of SAA drift in the future.

## 2. Instrumentation and Methods

### 2.1. CubeSat

The experimental data used for the preparation of this paper were measured by 1U CubeSat Lucky 7 (Figures 1 and 2). Lucky 7 (catalogue no. 19038W) was launched from Vostochny Cosmodrome by the Soyuz-2.1b rocket on 27 June 2019, to a quasi-synchronous orbit of inclination of 97.5008° and altitude of 520 km. Regular scientific data were collected until August 2019 after successful satellite testing. Examples of the points at which the radiation was measured are shown in Figure 3.

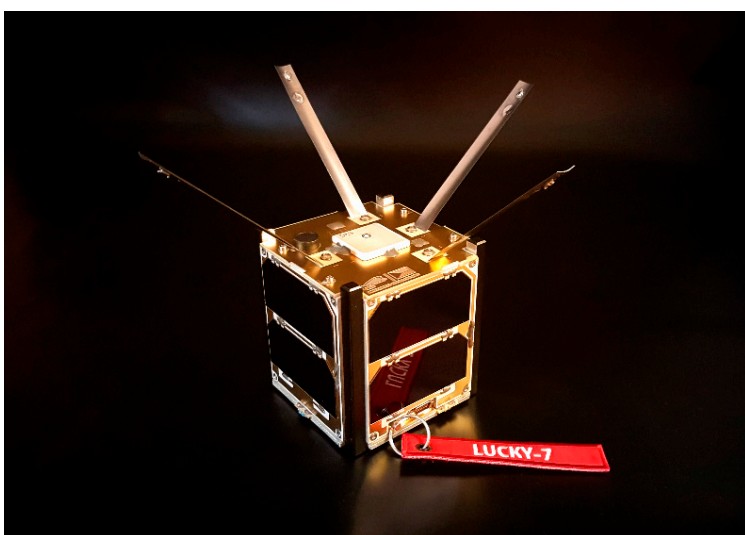

**Figure 1.** Lucky 7 CubeSat.

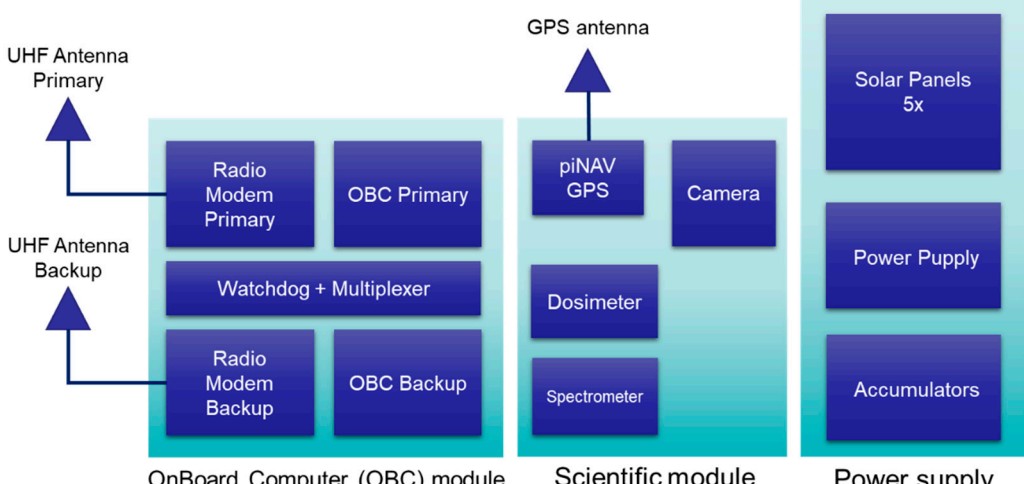

**Figure 2.** Lucky 7 block diagram.

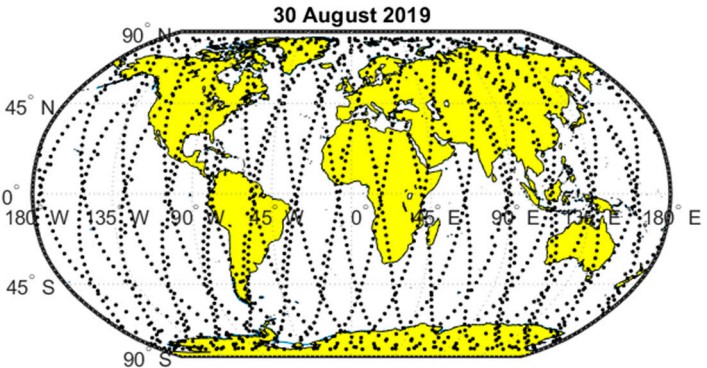

**Figure 3.** Typical points of radiation measurements.

Lucky 7 CubeSat is a private project of two people, namely Jaroslav Laifr and Pavel Kovar. Mr. Laifr developed and manufactured the satellite hardware including the power supply, radiation detector, and camera. He registered the satellite. Pavel Kovar developed the GPS receiver hardware and software and hardware and the software of the ultrahigh frequency (UHF) satellite modem, including a single-layer communication protocol capable of operating even at a high packet error rate. He is the author of the on-board computer hardware and software. He designed a ground station modem as well as ground station software.

The main component of the satellite is an onboard computer that integrates two independent computers and UHF modems that control the satellite, collect scientific data, and communicate with ground stations. The scientific module (Figure 4) is equipped with the GPS receiver piNAV 2 [32–34], the radiation detector piDOSE [35,36], a spectrometer and a low-resolution camera. The energy for satellite operation is generated by gallium arsenide solar cells that are mounted on the five satellite panels. The radiation hardening power supply is built from bipolar and silicon carbide transistors. Energy is stored in LiFe batteries with a capacity of 4 Wh.

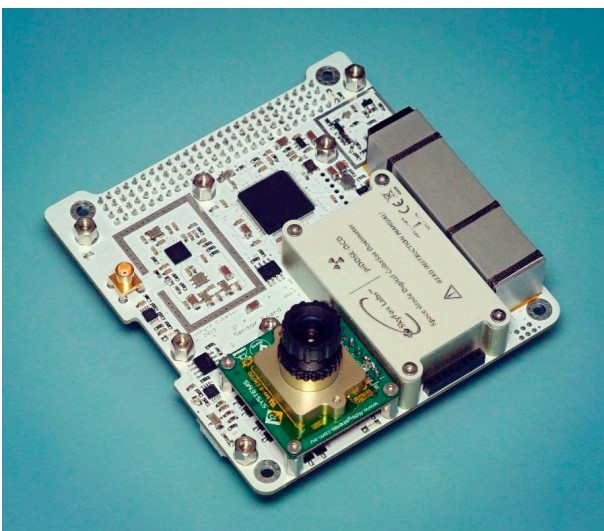

**Figure 4.** Scientific module.

The Lucky 7 satellite can realize measurements up to 2048 min in duration. The on-board computer saves to memory the number of pulses registered by a radiation detector, position and time of the end of the measurement, and satellite statuses such as power supply voltage, current, currents from the solar cells, the temperature of the detector, and other subsystems.

### 2.2. PiDOSE

The piDOSE radiation detector is based on a PIN diode X100-7 coupled with a CsI:Tl scintillator with dimensions of $4 \times 8 \times 8$ mm. The detector operates in particle counting mode. The energy deposition is not measured. The integration time is approximately 55 s. The data were converted to counts per minute and corrected for dead time. As shown in [36], the detector can register protons with energies higher than 30 MeV. Since the incident angle of particles and rotation of the satellite could not be determined, the counts were $4\pi$ normalized. Hence, isotropical irradiation is assumed. Although the irradiation in the SAA is strongly directional, the satellite slowly rotates, which helps to mitigate the different directional sensitivities of piDOSE. A typical observed radiation field is shown in Figure 5.

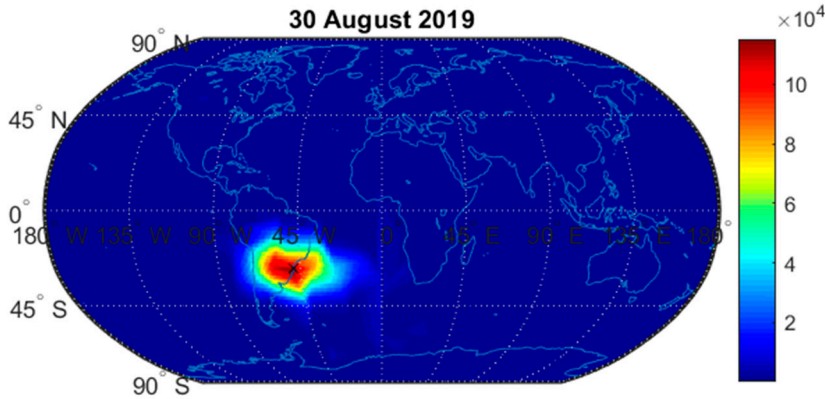

**Figure 5.** Example of the measurement of the radiation field (number of registered particles per minute (CMP)).

### 2.3. Position of the Measurement

As the satellite is not equipped with an altitude determination and control system, the satellite slowly rotates. The rotation rate is determined from the current of individual

solar panels or from the periodic variation in the carrier to noise (C/N0) indicator of the GPS receiver. The rotation rate is not constant; the typical value is one turn per minute [34]. This rotation causes the GPS position to be unavailable some of the time [34]. The typical availability of PNT (position, navigation, and timing) solutions is 80% of the time. The problem was solved by extrapolation of the satellite position.

The following proposed interpolation algorithm is based on the satellite motion equation and its modification for the Earth-centered Earth-fixed (ECEF) coordinate system [37,38]:

$$
\begin{aligned}
&\frac{dx}{dt} = V_x;\ \frac{dy}{dt} = V_y;\ \frac{dz}{dt} = V_z \\
&\frac{dV_x}{dt} = -\frac{\mu}{r^3}x - \frac{3}{2}J_0^2\frac{\mu \cdot a_e^2}{r^5}x\left(1 - \frac{5z^2}{r^2}\right) + \omega^2 x + 2\omega V_y \\
&\frac{dV_y}{dt} = -\frac{\mu}{r^3}y - \frac{3}{2}J_0^2\frac{\mu \cdot a_e^2}{r^5}y\left(1 - \frac{5z^2}{r^2}\right) + \omega^2 y + 2\omega V_x \\
&\frac{dV_x}{dt} = -\frac{\mu}{r^3}z - \frac{3}{2}J_0^2\frac{\mu \cdot a_e^2}{r^5}z\left(1 - \frac{5z^2}{r^2}\right)
\end{aligned}
\tag{1}
$$

where $(x, y, z)$ and $(V_x,\ V_y,\ V_z)$ are the position and velocity vectors, respectively, of the satellite in ECEF coordinates provided by the onboard GPS receiver; $r = \sqrt{x^2 + y^2 + z^2}$ is a satellite radius; $\mu = 398600.4 \times 10^9\ \text{m}^3/\text{s}^2$ is the standard gravitation parameter of the Earth; $J_0^2 = 1082625.7 \times 10^{-9}$ is the second-harmonic coefficient of geopotential; $\omega = 7.292115 \times 10^{-5}\ \text{rad/s}$ is the Earth rotation rate; and $a_e$ is the equatorial radius of the reference ellipsoid.

The satellite position is calculated by numerical integration of (1) using the fourth-order Runge–Kutta method [38].

The described algorithm enables the calculations of missing positions of the satellite and also enables the transformation of the position measurement from the end of the measurement cycle to its middle, as the satellite registers the position of the end of the measurement.

The ECEF position is, then, transformed to the geometrical coordinates LLH (longitude, latitude, and height) for further processing [38].

### 2.4. Position of the South Atlantic Anomaly (SAA)

The position of the SAA can be calculated as a centroid of the radiation field [13]. For this purpose, the data should be transformed to the latitude-longitude grid, and then, the centroid is calculated. The following formulas are presented based on [13]:

$$
\begin{aligned}
Latitude_{centroid} &= \frac{\sum(Interpolated\ Flux \times Flux's\ Latitude)}{\sum Interpolated\ Flux} \\
Longitude_{centroid} &= \frac{\sum(Interpolated\ Flux \times Flux's\ Longitude)}{\sum Interpolated\ Flux}
\end{aligned}
\tag{2}
$$

The problem with Equation (2) is that it does not consider the grid area; therefore, the individual grid points have equal mass. The problem is that the grid area decreases with latitude. Moreover, the method does not take into account that a one-degree grid is a spherical surface, and data are processed in two-dimension (2D). The proposed method is based on centroids in the space of constant curvature defined in [39]. The mass of the individual grids is calculated as a product of the number of registered particles per minute and grid area. The coordinates of the grid are transformed from the geodetic LLA coordinates to the ECEF, then the ECEF centroid coordinates are calculated using (3) and the results are transformed back to the LLH. The adjusted formulas for calculation of the longitudinal ECEF $(x_c, y_c, z_c)$ coordinates of the centroid adjusted to the grid area size are as follows:

$$
\begin{aligned}
x_c &= \frac{\sum(CPM(x,y,z)\cdot cos(\varphi)\cdot x)}{\sum CPM(x,y,z)} \\
y_c &= \frac{\sum(CPM(x,y,z)\cdot cos(\varphi)\cdot y)}{\sum CPM(x,y,z)} \\
z_c &= \frac{\sum(CPM(x,y,z)\cdot cos(\varphi)\cdot z)}{\sum CPM(x,y,z)}
\end{aligned}
\tag{3}
$$

where $\lambda$ and $\varphi$ are coordinates of the grid in LLH, $(x, y, z)$ are coordinates of the grid in ECEF, and $CPM(x, y, z)$ is the number of counts per minute in the grid.

### 2.5. Data Processing

The registered scientific data are downloaded by a ground station for further processing. One dataset contains data from approximately 22 orbits. The data processing can be summarized as follows:

(a) Calculation of the position of the satellite in the middle of the radiation detector, counting time by the method presented in Section 2.3.
(b) Resampling data to a one-degree grid.
(c) Calculation of the position of the radiation maximum as a centroid of the measured data (Section 2.4).
(d) Graphical presentation of the results.

## 3. Results

The radiation measurement in the SAA region is presented in Figure 6. The figure displays the development of the radiation field expressed as the number of particles registered per minute (CPM counts per minute) over the measurement campaigns, including the positions of the radiation maximum and centroid. The SAA positions are summarized in Table 2.

**Table 2.** Development of the position of the SAA.

| Measurement | Centroid Position Long. Lat. [°] | Max. Position Long. Lat. [°] |
|---|---|---|
| (a) 30 August 2019 | −25.8637 −48.6520 | −27 −49 |
| (b) 30 September 2019 | −26.1674 −48.4611 | −24 −48 |
| (c) 27 March 2020 | −26.2771 −49.1253 | −23 −50 |
| (d) 10 October 2020 | −26.4505 −48.8551 | −29 −55 |
| (e) 1 November 2020 | −26.4186 −49.2932 | −24 −53 |
| (f) 17 November 2020 | −26.5828 −48.7944 | −26 −57 |
| (g) 28 December 2020 | −26.1477 −48.5793 | −29 −53 |
| (h) 2 January 2021 | −26.6111 −49.0591 | −30 −60 |

Figure 7 displays the north–south and east–west cross-sections of the SAA radiation field measured in the individual campaigns.

The positions of the SAA radiation field maximum and centroid positions are also displayed in Figure 8. It is evident that the position of the maximum features a much higher scatter than the position of the centroid in which the scatter is considerably lower. The details of the centroid position scatter are shown in Figure 9.

The development of the position of the centroid was interpolated by a first-order polynomial (straight line), as shown in Figure 10. The resulting polynomials for latitude $\lambda_F$ and longitude $\varphi_F$ are as follows:

$$\lambda_F = -0.00093d - 26.0229 \qquad \varphi_F = -0.00062d - 48.65654 \tag{4}$$

where $d$ is the time in days from the first measurement (30 August 2019).

The position of the centroid is, therefore, moving by $0.00093°$ to the west and $0.00062°$ to the south per day, which is $0.34°$ to the west and $0.23°$ to the south per year.

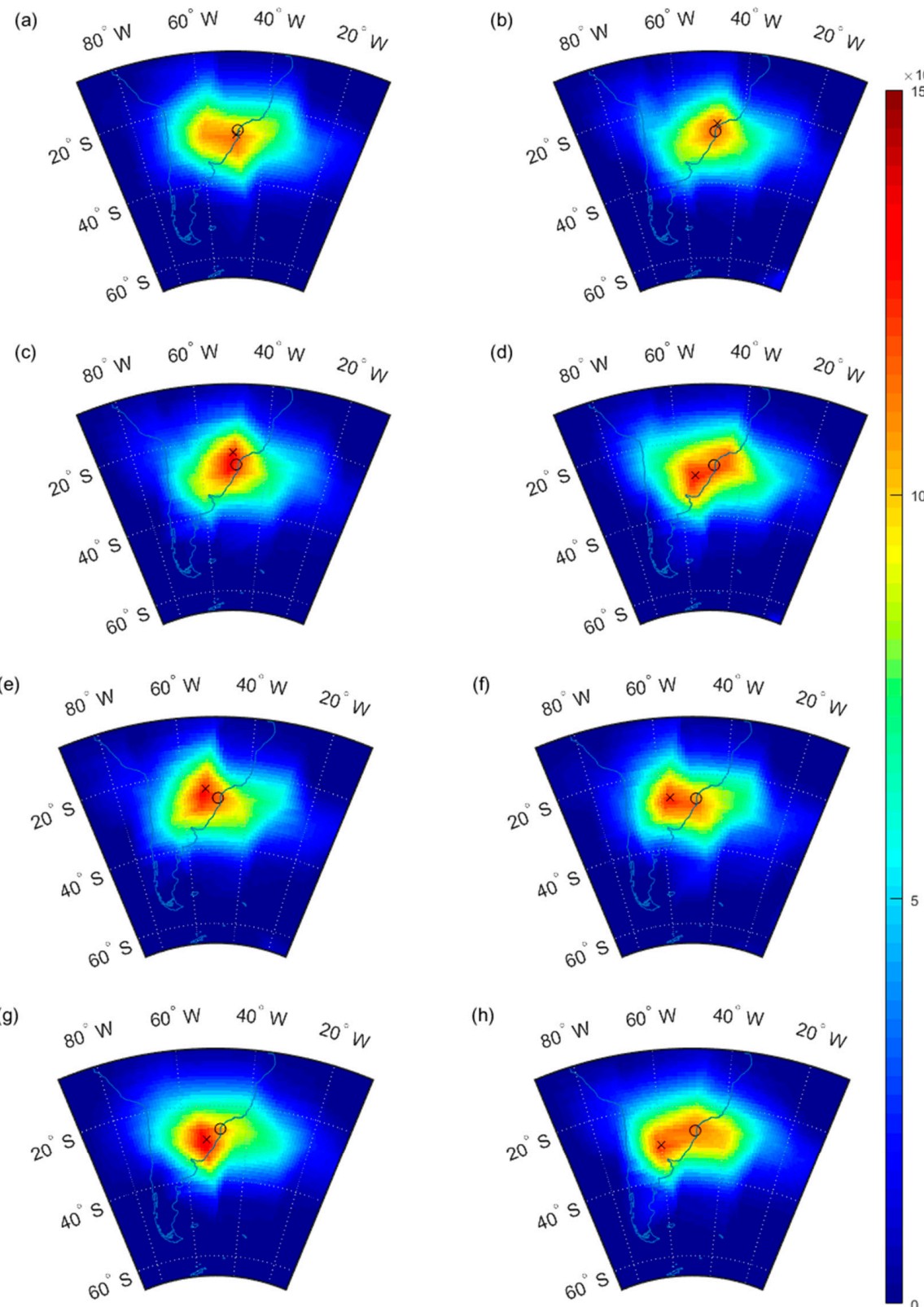

**Figure 6.** Development of the radiation field (CPM) in the SAA over measurements with the positions of the maximum of radiation (cross) and centroid (circle). (**a**) 30 August 2019; (**b**) 30 September 2019; (**c**) 27 March 2020; (**d**) 10 October 2020; (**e**) 1 November 2020; (**f**) 17 November 2020; (**g**) 28 December 2020; (**h**) 2 January 2021.

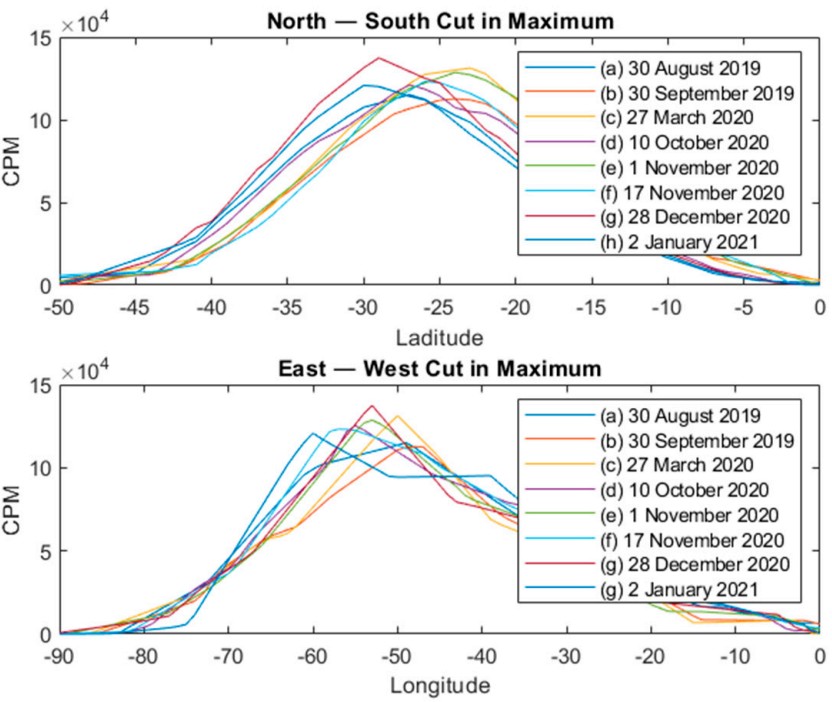

**Figure 7.** North–south and east–west cross-sections of the SAA radiation field.

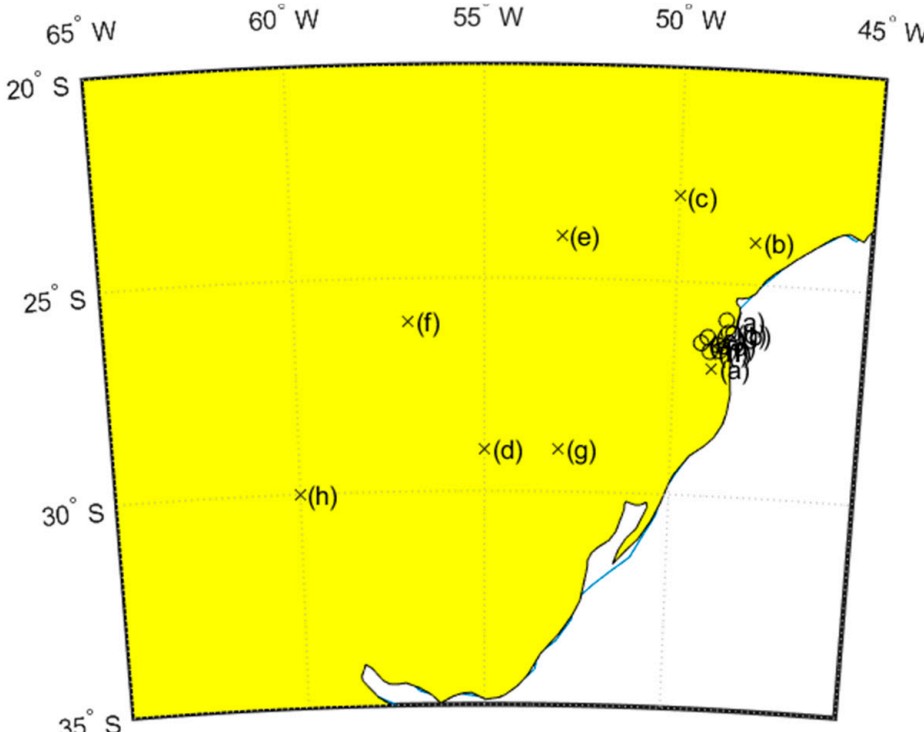

**Figure 8.** Development of the SAA maximum and centroid.

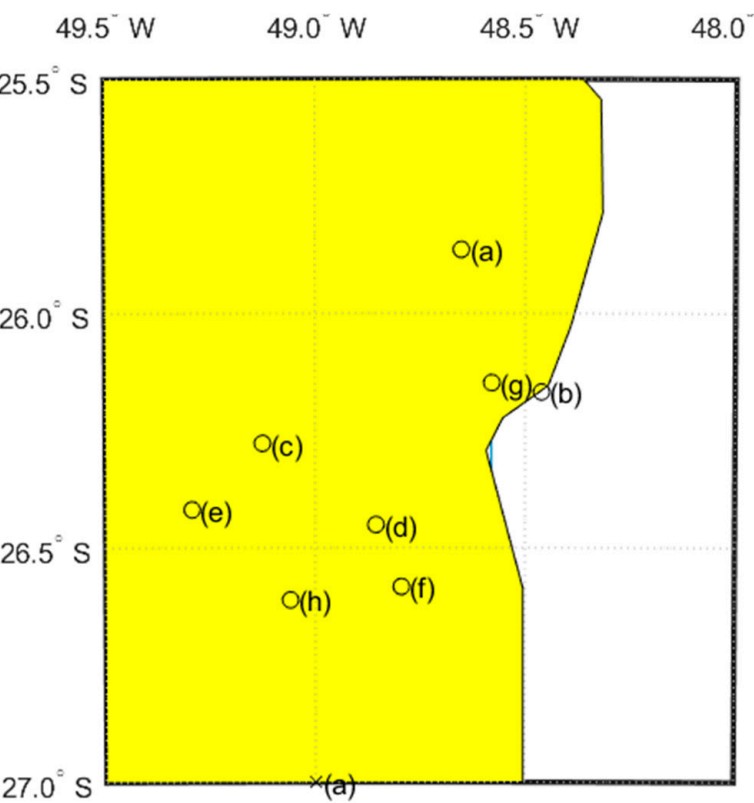

**Figure 9.** Development of the SAA centroid.

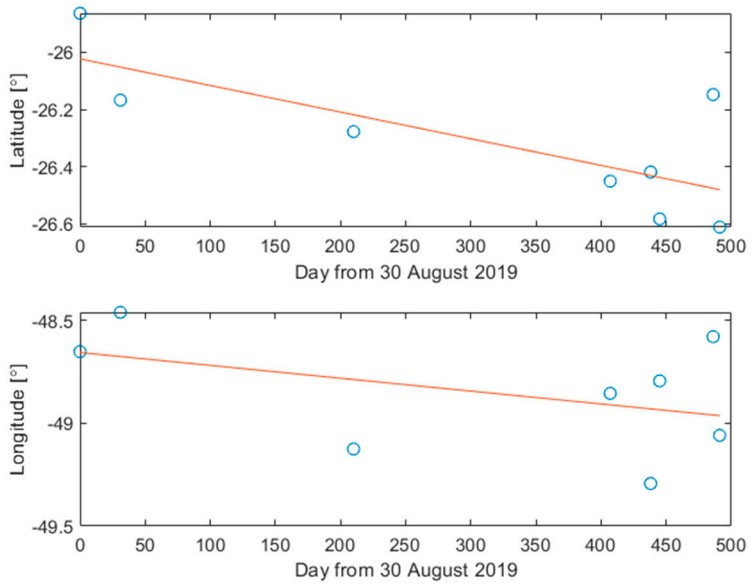

**Figure 10.** Straight line interpolation of the centroid position.

## 4. Discussion

The position of the South Atlantic Anomaly can be defined by various methods. The most intuitive is the position of the maximum of the fluence function. Alternatively, the position can be understood as a centroid of this function. In this paper, we applied both methods.

The position of the fluence maximum observed by individual measurement campaigns fluctuates more than the fluence centroid position. The positions of the fluence maxima are more sensitive than the position of the centroid to measurement noise. Although the

centroid method is relatively insensitive to the number of measurement points, it would benefit from denser sampling of the SAA region. Denser sampling can be obtained by selecting an orbit with higher inclination or by releasing a swarm of satellites.

Our results show the average westward movement of the SAA by 0.34°/year, which is in good agreement with previously reported measurements summarized in Table 1. The data show an average southward movement rate of 0.23°/year. Even though most of the publications show northward drift, southward drift was also observed in [2,3,21]. The inconsistency in the north–south direction can be explained by applying the improved centroid calculation method that correctly takes into account grid area, in contrast to the application of the less precise centroid position calculation method used in [13].

As shown in [2], the drift of the SAA in latitude is dependent on the energy range of the protons that are measured. Protons within the SAA with lower energy tend to drift towards the north, whereas the most energetic protons of the SAA move towards the south. In [34], piDOSE was estimated to be sensitive to protons with energies larger than 30 MeV due to the shielding of sensitive volume. Moreover, the latitudinal movement of the SAA depends on the solar cycle [21]. Often, there was a shift in direction during the solar cycle minima and maxima. Since our data captured the time of the solar cycle minimum, it can be expected to observe a similar shift in SAA movement. Another effect that influences the measurement of latitudinal movement is a systematic error caused by the fast latitudinal movement of a satellite. The particle counting time is about one minute. In this time, the satellite travels approximately four degrees in orbit which is projected mainly in the latitude. The maximal error in the determination of the latitude position of the fluence maximum within the counting time is two degrees. For the uniform distribution the mean systematic error is about one degree. The error in the east–west direction is considerably lower due to the slower movement of the satellite in longitude.

## 5. Conclusions

Our study has demonstrated that small, cost-effective 1U CubeSats can be successfully used for the observation of SAA drift. Unlike the measurements introduced in Table 1, which were performed on large and heavy-payload satellites, we used a simply designed detector made of commercial off-the-shelf components that was able to operate for more than 20 months in an increased-radiation environment.

The results show that the method using the fitting of maximum fluxes is suitable for evaluating the drift rate in a large dataset. However, the centroid method normalized for the cosine-latitude effect requires less data to reach the same accuracy. Hence, the latter method is more suitable for CubeSats that have limited performance and data transfer.

We believe that a swarm of 1U CubeSats equipped with similar detectors could be used as a system for continuously monitoring the movement of the SAA. The advantage of such an approach is that individual CubeSats of the swarm can be placed in different orbits and at more suitable inclinations that would allow denser sampling in the SAA region. This would significantly improve the coverage and would allow more complex analysis of SAA drift. Such information would be beneficial for designing more accurate models of the SAA. As shown by [20], on the one hand, the drift of the SAA can change suddenly due to "geomagnetic jerks", which are not incorporated into the models. On the other hand, several studies have shown strong periodicity in SAA movement [13,21] which can be anticipated and foreseen by the models. For example, the ESA's Space Environmental Information System (SPENVIS) [8] models an average drift of the SAA of 0.3° in the westward direction and no movement in latitude, although a number of studies have observed slight movements in the northward or southward direction.

**Author Contributions:** P.K. developed a satellite UHF communication system ground station, the onboard computer, the satellite software, and the piNAV GPS receiver; he also processed satellite data and drew figures and prepared the parts related to the Lucky 7 satellite; M.S. prepared the introduction and interpreted the experimental results. Both authors have read and agreed to the published version of the manuscript.

**Funding:** The data processing and preparation of this document were supported by the European Regional Development Fund, Project CRREAT no. CZ.02.1.01/0.0/0.0/15 003/0000481.

**Institutional Review Board Statement:** Not applicable.

**Informed Consent Statement:** Not applicable.

**Data Availability Statement:** The data that support the findings of this study are available from the author upon reasonable request.

**Acknowledgments:** The authors would like to thank Jaroslav Laifr for building the Lucky 7 CubeSat and developing the piDOSE radiation detector.

**Conflicts of Interest:** The authors declare no conflict of interest.

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
