# Peer review of "CubeSat Observation of the Radiation Field of the South Atlantic Anomaly"

_remotesensing, doi:10.3390/rs13071274_

Round 1

Reviewer 1 Report

Dear authors,

I cordially encourage scientific research conducted on nanosatellites. Your research appeared interesting. Let me summarize the points on how to make it thorough and sound. 

  1. It is important to properly cite the referenced work. Please check the contents of Table 1. I browsed the references and found that Badhwar[8] presents data for two short observation campaigns, whereas the Table shows 21.2 years (of observations?). Bühler 2002 [9] starts their abstract "During two years, from November 1994 to 1996", whereas the Table tells about 25 years. Konradi 1994 [7] describes their data in the following way: "The data presented in this study was derived from five shuttle missions which took place at various times over a period of two years". The Table lists 20 years for that research. I did not check the rest. 
  2. Line 84, subsection.
    1. As the satellite does not have an attitude determination system, how precisely might be the statement about slow rotation speed? Can you quantify what is slow? Without an attitude control, the rotation rate may be influenced by several torque sources. 
    2. What is PNT? It was never defined. 
    3. Is the position obtained from NORAD (North American Aerospace Defense Command) two-line elements not sufficient?
  3. Line 105, subsection. I agree with your challenge of research in Aubry et al. [18]. The equations (2) look simplistic. However, I would recommend calling a centroid a centroid. An example of calculations on a sphere is presented by G.A. Galperin, Commun. Math. Phys. 154, 63-84 (1993). You may also find useful the method in the Anderson et al. [17]
  4. Line 118. I can not see a logical sequence between minute-wise measurements and memory depth of 2048 points.
  5. Line 116, Data processing. Did you correct the count rate for the dead time of the detector? 
  6. The results section. Each experimental result inevitably comes with an error margin. They are vital for the section.
    1. Figures 12 and 13 will benefit from the presence of error bars or error regions. As the instrumental time resolution is 1 minute, each measurement averages about 4 degrees in orbit. It makes an impression of an important contribution to systematic error.
    2. Figure 14 reports an ambiguous message. Did you determine the position of the South Atlantic Anomaly by eight records, 22 orbits (~1.5 days) each?
    3. As the attitude of the spacecraft is not well known, could you correct for varying sensitivity the dosimeter has vs. incident direction?

Last but not least, the manuscript will deliver a better message if it is proof-read by an English native speaker.

Author Response

Reaction to reviewers’ comments

Dear reviewers, Dear editor-in-chief, thank you very much for the valuable comments, that enable us to improve the scientific value of our paper. The changes and reactions are written in red.

Reviewer 1.

  1. It is important to properly cite the referenced work. Please check the contents of Table 1. I browsed the references and found that Badhwar[8] presents data for two short observation campaigns, whereas the Table shows 21.2 years (of observations?). Bühler 2002 [9] starts their abstract "During two years, from November 1994 to 1996", whereas the Table tells about 25 years. Konradi 1994 [7] describes their data in the following way: "The data presented in this study was derived from five shuttle missions which took place at various times over a period of two years". The Table lists 20 years for that research. I did not check the rest. 

The time column shows the time that was used for averaging the SAA drift. In Badhwar it was a measurement of SKYLAB in December 1973 and MIR-18 in June 1995. In the case of Konradi the comparison is between AP-8 Max model 1970 with data from STS-31 in 1990. Bühler compared AP-8 model 1970 to data taken between 1994-1996. We have changed the content, so it shows the duration of the missions instead. The column is highlighted.

  1. Line 84, subsection.
    1. As the satellite does not have an attitude determination system, how precisely might be the statement about slow rotation speed? Can you quantify what is slow? Without an attitude control, the rotation rate may be influenced by several torque sources. 

We added explanation and reference:

The rotation rate is determined from the current of individual solar panels or from the periodic variation in the carrier to noise (C/N0) indicator of the GPS receiver. The rotation rate is not constant; the typical value is one turn per minute [34]. This rotation causes the GPS position to be unavailable some of the time [34].

    1. What is PNT? It was never defined. 

PNT is a standard abbreviation of position, navigation, and timing. We added an explanation to the text.

    1. Is the position obtained from NORAD (North American Aerospace Defense Command) two-line elements not sufficient?

The position and time of the measurement were measured by an on-board GPS receiver. The precision of the GPS receiver is about 10 m (95%). The precision of the NORAD elements is in the rank of hundreds of meters. No editing in the manuscript.

  1. Line 105, subsection. I agree with your challenge of research in Aubry et al. [18]. The equations (2) look simplistic. However, I would recommend calling a centroid a centroid. An example of calculations on a sphere is presented by G.A. Galperin, Commun. Math. Phys. 154, 63-84 (1993). You may also find useful the method in the Anderson et al. [17]

The reference has been added to the manuscript. We did the following change to clarify the problem.

The proposed method is based on centroids in the space of constant curvature defined in [39]. The mass of the individual grids is calculated as a product of the number of registered particles per minute and area.

The changes are highlighted.

  1. Line 118. I can not see a logical sequence between minute-wise measurements and memory depth of 2048 points.

For the simplicity reasons and the elimination of errors on the Onboard computer program, the satellite memory has fixed static organization. The content of the satellite memory is read by the ground station by a single-layer communication protocol. This enables to obtain and to interpret the data even if the packet error rate of the communication transceiver is very high. Fortunately, the communication system operates well and we are even able to routinely take snapshots and transmit them to the ground.

The satellite is built from the COTS components, not rad-hard ones. The satellite computer is therefore reset often, typically several times a day. The program is written to continue in the measurement after reset. When the computer is reset the time is lost, so the NORAD elements cannot be used, fortunately, the position and time are measured by GPS receiver, so that the time gap caused by the computer reset is not a problem.

The data from sensors are sampled once per minute. We also have a one-second mode of data acquisition, for example from the GPS receiver or solar cells, but this mode is not suitable for radiation sensors.

Due to reliability reasons, the sensors except GPS receiver are completely turn off and on at the end of the minute (power-on reset). The radiation sensor counting time therefore shorter. After the power on, the computer must send setup to the sensor, run the measurement, wait for measurement completion and receive the results. This shortens the radiation sensor counting time.

The counts from the radiation sensor recalculate onto the count per minute. We also take into account the mentioned delays for the calculation of the true position of the middle of the radiation sensor counting.

The paragraph was shifted to subsection 2.1. CubeSat. Change is highlighted.

  1. Line 116, Data processing. Did you correct the count rate for the dead time of the detector?

Yes, data were corrected for dead time. Please, see the previous answer.

  1. The results section. Each experimental result inevitably comes with an error margin. They are vital for the section.
    1. Figures 12 and 13 will benefit from the presence of error bars or error regions. As the instrumental time resolution is 1 minute, each measurement averages about 4 degrees in orbit. It makes an impression of an important contribution to systematic error.

Generally speaking, there is no problem to determine the function maximum from samples, if the Shannon sampling theorem is satisfied. But in the case of satellite measurements there besides the low dense sampling addition problem that values are measured at different times, because the fluence is not static, but is developing in time. This is a measurement methodology problem that can be solved by a swarm of the satellite, for example. A partial solution is to use orbit with the inclination that guarantees more dense sampling in longitude in shorter rime.

    1. Figure 14 reports an ambiguous message. Did you determine the position of the South Atlantic Anomaly by eight records, 22 orbits (~1.5 days) each?

Yes, more data are not available.

    1. As the attitude of the spacecraft is not well known, could you correct for varying sensitivity the dosimeter has vs. incident direction?

Since the rotation of the CubeSat is not known it is not possible to calculate the incident direction of particles. Hence no correction can be made. The 4pi normalization was used to average the error caused by different incident direction. This topic is discussed in piDOSE subsection of Instrumentations and Methods section.

Reviewer 2 Report

This paper describes the measurement of SAA radiation field by cubeseat. However, the paper is not written professionally, and there are many grammar errors and structure errors, which partly reflect that the preparation of the paper is pretty careless. Therefore, I recommend this paper to be resubmitted.

The author shall revised the format of the paper, those references are not in a bracket, which make the paper looks unprofessional

The abstract did not present the conclusion, please revise.

Line 21-23 change the sentence into ‘This creates region with reduced magnetic field intensity, which is called South Atlantic Anomaly (SAA) and located approximately at the east cost of Brazil.’

Line 26-28 the author shall add some references to support their argument that SAA is harmful.

Line 35 replace ‘As can be seen’ with ‘As is shown in previous studies, ’

Line 38 remove ‘,’ before ‘with very different’ and remove ‘very’ before ‘different instrumentation’

Line 64 replace ‘materials’ with ‘instrumentations’

Line 90 add bracket for the ECEF to show that it is the abbreviation of Earth Centered Earth Fixed

In line 105, there is the section 2.3, however, there is section 3.4 in line 116, is there missing content or just the error of section title??

Line 134 replace ‘be a Lucky 7’ with ‘by the Lucky 7’

In section 2, the figure is only to the Figure 3. However, in section 3, the figure is suddenly jumped to Figure 8. Combined with the problem with the error in section 3.4, I strongly doubt that there is a part of content missing in the paper.

Author Response

Reaction to reviewers’ comments

Dear reviewers, Dear editor-in-chief, thank you very much for the valuable comments, that enable us to improve the scientific value of our paper.

Reviewer 2.

This paper describes the measurement of SAA radiation field by cubeseat. However, the paper is not written professionally, and there are many grammar errors and structure errors, which partly reflect that the preparation of the paper is pretty careless. Therefore, I recommend this paper to be resubmitted.

The paper was revised and checked by professional English editing service. 

The author shall revised the format of the paper, those references are not in a bracket, which make the paper looks unprofessional

The references have been corrected.

The abstract did not present the conclusion, please revise.

The abstract has been revised and we added the missing information.

Line 21-23 change the sentence into ‘This creates region with reduced magnetic field intensity, which is called South Atlantic Anomaly (SAA) and located approximately at the east cost of Brazil.’

The proposed correction has been done.

Line 26-28 the author shall add some references to support their argument that SAA is harmful.

References have been added.

Line 35 replace ‘As can be seen’ with ‘As is shown in previous studies, ’

The proposed correction has been done.

Line 38 remove ‘,’ before ‘with very different’ and remove ‘very’ before ‘different instrumentation’

The proposed correction has been done.

Line 64 replace ‘materials’ with ‘instrumentations’

The proposed correction has been done.

Line 90 add bracket for the ECEF to show that it is the abbreviation of Earth Centered Earth Fixed

The proposed correction has been done.

In line 105, there is the section 2.3, however, there is section 3.4 in line 116, is there missing content or just the error of section title??

The numbering of sections has been changed.

Line 134 replace ‘be a Lucky 7’ with ‘by the Lucky 7’

The proposed correction has been done.

In section 2, the figure is only to the Figure 3. However, in section 3, the figure is suddenly jumped to Figure 8. Combined with the problem with the error in section 3.4, I strongly doubt that there is a part of content missing in the paper.

The numbering of figures has been changed.

Reviewer 3 Report

The authors describe a methodology of determining the location of the SAA using radiation measurements from a cubesat using two methods- using the location of the maximum fluence of the radiation and by using the centroid of the radiation field around the SAA. In my view, it is unclear what is the novelty of the approach other than a modest improvement in the addition of the cosine-latitude effect to compute the location of the SAA. The quality of the paper certainly needs improvement. The authors may consider my comments and questions below to improve the quality before publication.

  1. The authors need to proofread the paper to address several typographical errors throughout the paper. I will only highlight a few major ones.
  2. The authors should include a little more detail in the abstract to provide more context about the work and the uniqueness of their approach.
  3. The authors should emphasize what makes their measurement concept or algorithm unique. It appears as though these measurements are not unique as evidenced by the number of publications that have preceded this one. Perhaps the authors propose a more cost-effective approach using cubesats? If so, this needs to be highlighted.
  4. In Sec. 2.1., the description of the Cubesat, the authors may want to include details about who/where the instrument was built, and perhaps other relevant details.
  5. Line 69, the cubesat is placed in close to sun-synchronous orbit which enables global coverage. If the target site is well known - the South Atlantic Anomaly, are there other orbits that are better suited for gaining better sampling of that specific area? 
  6. Sec. 2.2., the first paragraph needs to be improved. It is unclear what the authors intended with the word 'souses'. Also, the term 'PNT' is first mentioned here without any explanation. 
  7. In Equation 1, the term a_e is not defined.
  8. Line 104, the word 'High' needs to be replaced with 'Height'.
  9. Line 118, the authors use the word 'therefore' to indicate that it is self-evident that the measurement time for radiation detector is 1 minute. It does not appear to be obvious unless I am missing some detail.
  10. Line 118, I interpret the measurement time of the radiation detector to be the 'integration time'. Are the radiation measurements from the dosimeter analog or digital? What is the sampling rate for this detector? 
  11. Line 124, the authors use the term 'rounds'. Do they intend to refer to the number of orbits? If so, please make the clarification/correction.
  12. In Figures 9 and 10, what are the units of the radiation measurement? If these are number of particles, these should be written in the legend or indicated some place else.
  13. Figure 11, are these meant to represent cross- sections of the radiation fields at the SAA? What does the label 'CPM' on the y-axis mean?
  14. Lines 203-213, the authors make a case for a constellation to measure the SAA at a higher temporal resolution. Could the authors estimate at what temporal resolution would these need to be measured at? Does the modeling of the movement of the SAA seem to provide reasonable estimates for predicting the location of the SAA? The authors may want to consider my comment No. 5 and think about how they may want to approach this in the future. 

Author Response

Reaction to reviewers’ comments

Dear reviewers, Dear editor-in-chief, thank you very much for the valuable comments, that enable us to improve the scientific value of our paper. 

Reviewer 3.

The authors describe a methodology of determining the location of the SAA using radiation measurements from a cubesat using two methods- using the location of the maximum fluence of the radiation and by using the centroid of the radiation field around the SAA. In my view, it is unclear what is the novelty of the approach other than a modest improvement in the addition of the cosine-latitude effect to compute the location of the SAA. The quality of the paper certainly needs improvement. The authors may consider my comments and questions below to improve the quality before publication.

  1. The authors need to proofread the paper to address several typographical errors throughout the paper. I will only highlight a few major ones.

The proofread has been done by professionals. The certificate is attached.

  1. The authors should include a little more detail in the abstract to provide more context about the work and the uniqueness of their approach.

The abstract has been extended so it reflects the uniqueness of our work.

  1. The authors should emphasize what makes their measurement concept or algorithm unique. It appears as though these measurements are not unique as evidenced by the number of publications that have preceded this one. Perhaps the authors propose a more cost-effective approach using cubesats? If so, this needs to be highlighted.

The uniqueness of our work is mainly in presentation of data on SAA drift which were taken by small unexpensive 1U CubeSat unlike the data presented in the past which were obtained by large and heavy satellites. Our results show that the monitoring of SAA drift can be done by small cost-effective CubeSats. As far as we know the SAA drift data measured by CubeSat has never been published although there has been several CubeSats missions with much more advanced radiation detectors which probably could have reported on SAA drift. They are mentioned in the Introduction. Moreover, the presented modified centroid method offers the accuracy improvement for calculation of SAA centroid. We have modified the text, so it emphasizes the uniqueness.

  1. In Sec. 2.1., the description of the Cubesat, the authors may want to include details about who/where the instrument was built, and perhaps other relevant details.

We added the following explanation to the manuscript:

Lucky 7 CubeSat is a private project of two people, namely, Jaroslav Laifr and Pavel Kovar. Mr. Laifr developed and manufactured satellite hardware including the power supply, radiation detector, and camera. He registered the satellite. Pavel Kovar developed GPS receiver hardware and software and hardware and software of the ultrahigh frequency (UHF) satellite modem, including a single-layer communication protocol capable of operating even at a high packet error rate. He is the author of the on-board computer hardware and software. He designed a ground station modem as well as ground station software.

  1. Line 69, the cubesat is placed in close to sun-synchronous orbit which enables global coverage. If the target site is well known - the South Atlantic Anomaly, are there other orbits that are better suited for gaining better sampling of that specific area? 

We can imagine some orbit with a more suitable inclination that guarantees denser scanning in longitude, but the authors could not select the orbit. The main criterion was the low launch price and availability for private customers.

  1. Sec. 2.2., the first paragraph needs to be improved. It is unclear what the authors intended with the word 'souses'. Also, the term 'PNT' is first mentioned here without any explanation. 

The same problem was identified by the first reviewer. The problem was solved.

  1. In Equation 1, the term a_e is not defined.

ae is the equatorial radius of the reference ellipsoid. We added an explanation to the manuscript.

  1. Line 104, the word 'High' needs to be replaced with 'Height'.

We did the proposed correction.

  1. Line 118, the authors use the word 'therefore' to indicate that it is self-evident that the measurement time for radiation detector is 1 minute. It does not appear to be obvious unless I am missing some detail.

A detailed explanation of the problem is in answer to reviewer 1.

  1. Line 118, I interpret the measurement time of the radiation detector to be the 'integration time'. Are the radiation measurements from the dosimeter analog or digital? What is the sampling rate for this detector? 

The paragraph was clarify and shifted to the ceubsection 2.1. CubeSat

  1. Line 124, the authors use the term 'rounds'. Do they intend to refer to the number of orbits? If so, please make the clarification/correction.

We replaced rounds with orbits.

  1. In Figures 9 and 10, what are the units of the radiation measurement? If these are number of particles, these should be written in the legend or indicated some place else.

Figures 9 and 10 display the number of registered particles per minute (CPM). We added this information to the legends.

  1. Figure 11, are these meant to represent cross- sections of the radiation fields at the SAA? What does the label 'CPM' on the y-axis mean?

CPM means counts per minute. We added an explanation to the text.

  1. Lines 203-213, the authors make a case for a constellation to measure the SAA at a higher temporal resolution. Could the authors estimate at what temporal resolution would these need to be measured at? Does the modeling of the movement of the SAA seem to provide reasonable estimates for predicting the location of the SAA? The authors may want to consider my comment No. 5 and think about how they may want to approach this in the future. 

The discussion about modelling thee SAA movement as well as the future approach was added to the Discussion section.

Reviewer 4 Report

Dear authors and editor,

The manuscript reports an application of CubeSat to determine the South Atlantic Anomaly's radiation field. The topic may be of interest, but the quality of the manuscript needs to be improved. The Introduction probably needs a section on the use of CubeSat in other practical applications. Moreover, the objectives of the study are not specified. M&M report very well the technical approach, but a description of the experimental design adopted by the authors is missing. For example, the observation time is presented only in the Results section. The Discussion and the Conclusions are very short and should be developed to highlight the advantages and disadvantages of the proposed methodology. English editing must be improved.

Here are some specific comments:

References: when more than two references are reported, they should be listed within the same brackets, i.e. [1,….9]

Line 75: provide the definition of the acronym UHF

Line 87: provide the definition of the acronym PNT

Lines 133-140: I believe these lines should be moved to the M&M section

Author Response

Reaction to reviewers’ comments

Dear reviewers, Dear editor-in-chief, thank you very much for the valuable comments, that enable us to improve the scientific value of our paper. 

Reviewer 4.

The manuscript reports an application of CubeSat to determine the South Atlantic Anomaly's radiation field. The topic may be of interest, but the quality of the manuscript needs to be improved. The Introduction probably needs a section on the use of CubeSat in other practical applications. Moreover, the objectives of the study are not specified. M&M report very well the technical approach, but a description of the experimental design adopted by the authors is missing. For example, the observation time is presented only in the Results section. The Discussion and the Conclusions are very short and should be developed to highlight the advantages and disadvantages of the proposed methodology. English editing must be improved.

 The paragraph on CubeSats has been added to the Introduction section. The mission objectives have been added to the Introduction section. The details on piDOSE detector has been added to the section Instrumentations and Methods. The Discussion and Conclusions have been extended.

Here are some specific comments:

References: when more than two references are reported, they should be listed within the same brackets, i.e. [1,….9]

 The references have been changed.

Line 75: provide the definition of the acronym UHF

UHF means Ultra High Frequency. We added an explanation to the manuscript.

Line 87: provide the definition of the acronym PNT

The same problem was identified by the first reviewer. The problem was solved.

Lines 133-140: I believe these lines should be moved to the M&M section

The lines were moved to section Instrumentation and Methods.

Round 2

Reviewer 1 Report

Dear authors, thank you for your time and consideration. The manuscript has improved. I must highlight the points that could be improved.

Is the position obtained from NORAD (North American Aerospace Defense
Command) two-line elements not sufficient?

The position and time of the measurement were measured by an on-board GPS receiver. The precision of the GPS receiver is about 10 m (95%). The precision of the NORAD elements is in the rank of hundreds of meters. No editing in the manuscript.

I have asked whether you find it sufficient or not with argumentation. I am aware that GPS-based location is more precise. I question the benefit of the accuracy given that the count rate is averaged over a four-degree arc. 

Line 105

The proposed method is based on centroids in the space of constant curvature defined in [39]. The mass of the individual grids is calculated as a product of the number of registered particles per minute and area.

Are you sure that the method in [39] and the proposed method are effectively the same? The method you propose is a cartesian averaging on the rectangular grid, which is the equirectangular projection of a sphere. 

Figures 12 and 13

Generally speaking, there is no problem to determine the function maximum from samples, if the Shannon sampling theorem is satisfied. But in the case of satellite measurements there besides the low dense sampling addition problem that values are measured at different times, because the fluence is not static, but is developing in time. This is a measurement methodology problem that can be solved by a swarm of the satellite, for example. A partial solution is to use orbit with the inclination that guarantees more dense sampling in longitude in shorter rime.

I agree that the issue can be solved with a swarm of satellites. However, let us consider the presented research. The Shannon theorem does not allow for high precision at a given interpolated point unless it is well established the input signal spectrum has been limited, e.g., by a low-pass filter. A higher frequency harmonic would be interpreted incorrectly as a low-frequency one (aliasing).

Your data are 4-degree stripes with a constant (averaged over one minute) intensity. There is no strong indication of where is the centroid of each stripe. In other words, centroids of each segment are not in their segments' midpoints. Therefore, I suggest you consider the systematic error presented by each 4-degree measurement. 

The experimental data look scarce, but I am positive that a proper elaboration on the methodology can make a good article.

Author Response

Reaction to reviewers’ comments

Dear reviewers, Dear editor-in-chief, thank you very much for the valuable comments, that enable us to improve the scientific value of our paper. The changes and reactions are written in red.

Reviewer 1.

Dear authors, thank you for your time and consideration. The manuscript has improved. I must highlight the points that could be improved.

Is the position obtained from NORAD (North American Aerospace Defense
Command) two-line elements not sufficient?

The position and time of the measurement were measured by an on-board GPS receiver. The precision of the GPS receiver is about 10 m (95%). The precision of the NORAD elements is in the rank of hundreds of meters. No editing in the manuscript.

I have asked whether you find it sufficient or not with argumentation. I am aware that GPS-based location is more precise. I question the benefit of the accuracy given that the count rate is averaged over a four-degree arc. 

Authors’ reply

You are right, the NORAD elements position precision would be sufficient for our measurement. We need only time from the GPS, because the on-board computer is not capable to keep time in the case of reset. The computer is built from the COTS component and resets comes several times a day.

The ideal sampling function is a Dirac distribution, for band-limited signals, it is sin(x)/x (sampling) function. The rectangular counting brings some systematic error to the measurement because the spectrum of the rectangular function is infinite in the ideal case, so the Shannon theorem is not fulfilled to one hundred percent. This is a systematic error, that cannot be solved at this time. No changes in the manuscript.

Line 105

The proposed method is based on centroids in the space of constant curvature defined in [39]. The mass of the individual grids is calculated as a product of the number of registered particles per minute and area.

Are you sure that the method in [39] and the proposed method are effectively the same? The method you propose is a cartesian averaging on the rectangular grid, which is the equirectangular projection of a sphere.

Authors’ reply 

Thank you very much for clarification of the comment. We modified the algorithm from

Current algorithm

  1. Transformation of the position from ECEF to the LLH.
  2. Resampling the measurement to the one-degree grid
  3. Calculation of the mass of the grid
  4. Calculation of the centroid in longitude, latitude grid

To

Modified algorithm

  1. Transformation of the position from ECEF to the LLH.
  2. Resampling the measurement to the one-degree grid
  3. Calculation of the mass of the grid
  4. Transformation of the resampled data from LLH to the ECEF
  5. Calculation of the centroid in ECEF (3D)
  6. Transformation of the results to the LLH coordinates

We modified paragraph 2.3, equation (3), updated table 2 and Figures 6, 8, 9, equation (4) and discussion and conclusion sections 

Figures 12 and 13

Generally speaking, there is no problem to determine the function maximum from samples, if the Shannon sampling theorem is satisfied. But in the case of satellite measurements there besides the low dense sampling addition problem that values are measured at different times, because the fluence is not static, but is developing in time. This is a measurement methodology problem that can be solved by a swarm of the satellite, for example. A partial solution is to use orbit with the inclination that guarantees more dense sampling in longitude in shorter rime.

I agree that the issue can be solved with a swarm of satellites. However, let us consider the presented research. The Shannon theorem does not allow for high precision at a given interpolated point unless it is well established the input signal spectrum has been limited, e.g., by a low-pass filter. A higher frequency harmonic would be interpreted incorrectly as a low-frequency one (aliasing).

Your data are 4-degree stripes with a constant (averaged over one minute) intensity. There is no strong indication of where is the centroid of each stripe. In other words, centroids of each segment are not in their segments' midpoints. Therefore, I suggest you consider the systematic error presented by each 4-degree measurement. 

Authors’ reply

The comment is absolutely right. The counting of the duration of one minute corresponds to the 4-degree shift of the satellite in the orbit. As the satellite was placed in a nearly polar orbit (inclination 97.6 degrees), this movement is projected mainly to the latitude. The worst-case latitude error of the position of the radiation maximum is then two degrees. If the error distribution is uniform, the mean value of the error is one degree. The pessimistic estimation of the error mean value is, therefore, one degree.

We added this discussion to the conclusion. Corrections are highlighted.    

The experimental data look scarce, but I am positive that a proper elaboration on the methodology can make a good article.

Reviewer 2 Report

The author has made necessary revisions to improve the paper, and now all my concerns are solved. I recommend this paper to be published in present form

Author Response

Reaction to reviewers’ comments

Dear reviewers, Dear editor-in-chief, thank you very much for the valuable comments, that enable us to improve the scientific value of our paper. The changes and reactions are written in red.

Reviewer 2.

No question to be resolved.

Reviewer 3 Report

The authors have adequately addressed my concerns. I think the paper is much improved and is now appropriate for publication. 

Author Response

Reaction to reviewers’ comments

Dear reviewers, Dear editor-in-chief, thank you very much for the valuable comments, that enable us to improve the scientific value of our paper. The changes and reactions are written in red.

Reviewer 3.

No question to be resolved.

Reviewer 4 Report

Dear Editor snd Authors,

undoubtedly, the authors addressed my comments and the scientific sound of the manuscript has been improved. I still have the following concerns before accepting the manuscript for publication:

  • when I've asked for the objectives, I meant the objectives of this study and not the objectives of CubeSats. Therefore, the objectives still need to be added
  • English editing must be improved

Best regards

Author Response

Reaction to reviewers’ comments

Dear reviewers, Dear editor-in-chief, thank you very much for the valuable comments, that enable us to improve the scientific value of our paper. The changes and reactions are written in red.

Reviewer 4.

Dear Editor snd Authors,

undoubtedly, the authors addressed my comments and the scientific sound of the manuscript has been improved. I still have the following concerns before accepting the manuscript for publication:

  • when I've asked for the objectives, I meant the objectives of this study and not the objectives of CubeSats. Therefore, the objectives still need to be added
  • English editing must be improved

Best regards

A paragraph on the objectives of the study was added to the introduction.

Round 3

Reviewer 1 Report

Dear authors, thank you for the work on the article. I am satisfied with your answers.